# A Comparison of Ethanol, Methanol, and Butanol Blending with Gasoline and Its Effect on Engine Performance and Emissions Using Engine Simulation

Simeon Iliev

Department of Engines and Vehicle, University of Ruse, 8 Studentska Street, 7017 Ruse, Bulgaria; spi@uni-ruse.bg

**Abstract:** Air pollution, especially in large cities around the world, is associated with serious problems both with people's health and the environment. Over the past few years, there has been a particularly intensive demand for alternatives to fossil fuels, because when they are burned, substances that pollute the environment are released. In addition to the smoke from fuels burned for heating and harmful emissions that industrial installations release, the exhaust emissions of vehicles create a large share of the fossil fuel pollution. Alternative fuels, known as non-conventional and advanced fuels, are derived from resources other than fossil fuels. Because alcoholic fuels have several physical and propellant properties similar to those of gasoline, they can be considered as one of the alternative fuels. Alcoholic fuels or alcohol-blended fuels may be used in gasoline engines to reduce exhaust emissions. This study aimed to develop a gasoline engine model to predict the influence of different types of alcohol-blended fuels on performance and emissions. For the purpose of this study, the AVL Boost software was used to analyse characteristics of the gasoline engine when operating with different mixtures of ethanol, methanol, butanol, and gasoline (by volume). Results obtained from different fuel blends showed that when alcohol blends were used, brake power decreased and the brake specific fuel consumption increased compared to when using gasoline, and CO and HC concentrations decreased as the fuel blends percentage increased.

**Keywords:** alcohols; methanol; ethanol; butanol; emissions





## 1. Introduction

The depletion of crude oil is a problem that has arisen in recent decades [1]. Numerous studies have been conducted in recent years to find substitutes for fossil fuels [2,3]. Another very important problem is the combustion gases emitted from ICE (internal combustion engines) that negatively impact nature and human health [4]. The main harmful exhaust gases released from ICE are hydrocarbons (HC), nitrogen oxides (NOx), carbon monoxide (CO), and pollutants from particles [5]. $CO_2$ emissions are estimated as a pollutant because they cause global warming. In addition, $CO_2$ is contributes to greenhouse gas emissions (GHGE) [6]. $CO_2$ is dangerous for people because it takes the place of oxygen in the air people breathe; it is not toxic, but it causes hypoxia in the body [7]. To be able to use ICE with zero $CO_2$ emissions it is necessary to use carbon-free fuels or carbon fuels with zero GHG emissions. Fuels meeting these requirements are fuels produced from biomass or captured $CO_2$ [8]. Alternative fuels or energy sources are obtained from resources other than petroleum, which can contribute to the decarbonisation of transport and improve the environmental performance of the transport sector. These fuels are also renewable [9,10]. The fuels that are most often used as alternative fuels are propane [11], natural gas [12], methanol [13], ethanol [14], butanol [15], and hydrogen [16]. Many of these fuels are used as additives to fossil fuels (gasoline and diesel fuels) and can be blended with them. Fuel additives can be added to the base fuel to improve its properties [17]. Alcoholic fuels, which can be mixed with fossil fuels and used in ICE, play the role of an alternative fuel [18]. Several oxygen-containing fuels are used as fuel blends, such as butanol, ethanol,

methanol, and others [19]. With the use of oxygenated fuel blends, more oxygen enters the combustion chamber, and their use in ICE can reduce engine emissions [20]. One positive characteristic of alcohols as liquid fuels is that they can operate in engines without the need for substantial modification to the fuel system. The alcohols such as methanol, ethanol, and butanol can be used both in a clean form and in a blended form with diesel or gasoline to reduce demand for conventional fuels [21].

### 1.1. Properties of Methanol

Methanol can be produced from coal, biomass, or even natural gas with acceptable energy cost. The chemical and physical properties of the methanol used in internal combustion engines produce low emissions. Thanks to being less reactive than gasoline, its evaporative emissions contribute less to smog formation. Methanol also has a latent heat of evaporation (Table 1), which results in lower combustion temperatures, resulting in a decrease in NOx formation [22,23].

**Table 1.** Comparison of fuel properties of gasoline, methanol, ethanol, and butanol [24–26].

| Properties | Gasoline | Methanol | Ethanol | Butanol |
|---|---|---|---|---|
| Chemical formula | $C_8H_{15}$ | $CH_3OH$ | $C_2H_5OH$ | $C_4H_9OH$ |
| Molar mass, kg/kmol | 114 | 32 | 46 | 74.12 |
| Oxygen content, wt% | - | 50 | 34.73 | 21.59 |
| Carbon content, wt% | 86 | 38 | 52 | 65 |
| Hydrogen content, wt% | 14 | 12 | 13.1 | 13.5 |
| Stoichiometric AFR | 14.5 | 6.43 | 8.94 | 11.12 |
| Lower heating value, MJ/kg | 44.3 | 20.1 | 27 | 33 |
| Higher heating value, MJ/kg | 48 | 22.88 | 29.85 | 36.07 |
| Volumetric energy content, MJ/m$^3$ | 31746 | 15871 | 21291 | 26795 |
| Heat of evaporation, kJ/kg at 1 bar | 375 | 1089 | 841 | 584 |
| Research octane number | 96.5 | 112 | 111 | 96 |
| Motor octane number | 87.2 | 91 | 92 | 81 |
| Cetane number | - | <5 | 8 | 17–25 |
| Boiling temperature, °C at 1 bar | 25–215 | 65 | 79 | 118 |
| Vapor pressure, bar at 20 °C | 0.25–0.45 | 0.13 | 0.059 | 0.064 |
| Critical pressure, bar | - | 81 | 63 | 45 |
| Critical temperature, °C | - | 239.4 | 241 | 290 |
| Kinematic viscosity, cSt at 20 °C | 0.6 | 0.74 | 1.2 | 3.6 |
| Density, kg/cm$^3$ | 740 | 798 | 785 | 811 |
| Surface tension, mN/m at 20 °C | 21.6 | 22.1 | 22.3 | 24.57 |
| Minimum ignition energy, mJ at $\varphi = 1$ | 0.8 | 0.21 | 0.65 | - |
| Auto-ignition temperature, °C | 192–470 | 465 | 425 | 343 |
| Peak flame temperature, °C at 1 bar | 2030 | 1890 | 1.920 | - |
| Adiabatic flame temperature, K | ~2275 | 2143 | 2193 | 2262 |
| Flammability limits, vol% | 1.4–7.6 | 6–36 | 3–19 | 1.7–12 |
| Flash point, °C | −45 | 12 | 14 | 35 |
| Bulk modulus, N/mm$^2$ at 20 °C 2 MPa | 1300 | 823 | 902 | - |
| Specific $CO_2$ emissions, g/MJ | 73.95 | 68.44 | 70.99 | 71.9 |
| Specific $CO_2$ emissions relative to gasoline | 1 | 0.93 | 0.96 | 0.97 |

Emissions generated by the evaporation of methanol during transport, storage, disposal, and use are half of those with gasoline but increase with the use of gasoline/methanol mixtures. Due to its lower calorific value, almost twice as much methanol by volume is necessary to achieve an equivalent power as that of benzene, but the evaporation losses of methanol may be about two-thirds those of gasoline. During combustion, unburned fuel hydrocarbons (unburned fuel) is less reactive because it is primary methanol. Due to the lower specific reactivity of methanol, unburned methanol and evaporated methanol emissions are less likely to form smog/ozone than are an equal weight of organic emissions from petrol engines.

Since methanol does not contain sulphur, its use as fuel contributes to the reduction of sulphur dioxide ($SO_2$). The use of methanol will also help reduce acid rain, as SO and NOx emissions lead to acidic acid deposition. At the same fuel efficiency, the $CO_2$ emissions emitted by vehicles working with methanol are theoretically around 94% of the emissions of similar vehicles with petroleum. The production of methanol (derived by steam reformation of natural gas) releases almost half of the greenhouse gases as those released to produce gasoline. When the entire fuel cycle of the fuel resource is included, methanol has very similar greenhouse gas emissions as those of gasoline. Spark-ignition engines using methanol as fuel are 15–20% more efficient than those using gasoline [27,28]. This is due to lean-burn technology (usually used in modern direct injection engines) that is possible through the wide flammable limits of methanol. The engines with lean-burn technology are characterized by better thermal efficiency, lower exhaust emissions, and simpler oxidation and catalyst technology. HC and CO emissions have been shown to be much lower than those released by other fuels, while $NO_X$ emissions are approximately the same as those of current petrol vehicles. Liu et al. [29] conducted a study with a spark ignition port fuel injection engine and methanol/gasoline blends of 0%, 15%, 20%, 25%, and 30% of methanol in volume (M5, M10, and M15) under full load conditions (without any changes of the parameters of engine). Their study showed the power, torque, CO, and HC of the tested engine fuelled with blended fuels were decreased. Geng et al. [30] studied a PFI engine with three different ratios of methanol/gasoline blends of 0%, 15%, and 45% of methanol in volume (M0, M15, and M45). Their research showed that cylinder gas pressure and heat release rate occurred earlier and increased with the increase of methanol concentrations in mixtures. Particulate number (PN) and mass concentration also decreased with low concentrations of methanol/gasoline while it increased significantly in blends with higher proportions of methanol to gasoline. Agarwal et al. [31] studied the performance and emissions of methanol/gasoline blends (10% and 20%) and compared them with net gasoline in a gasoline engine. Their experiments were conducted under a partial load condition. The experimental results showed that methanol/gasoline blends increased brake thermal efficiency and lowered the emissions of NO, CO, and smoke.

### 1.2. Properties of Ethanol

Ethanol is considered to be one of the prospective fuels for use in ICE because it can be produced from waste materials or natural products, unlike gasoline and diesel fuels, which are non-renewable [32,33]. If any comparison is to be made regarding the renewability of alcohols, ethanol surpasses methanol because ethanol can be manufactured by alcoholic fermentation of biomass feedstocks (corn, sugarcane), while methanol basically produced from petrol fuels or coal. Another important characteristic of ethanol is that it can be used without any major changes in the fuel system of the spark-ignited engines. Ethanol is known as the best suited fuel for gasoline engines among the various alcohols. Ethanol can be used as a transportation fuel in the following three ways:

- Directly used as a fuel or with 15% or more gasoline, known as "E85". It can also be directly used in diesel engines specially configured for alcoholic fuels;
- As "gasohol", blended with gasoline, usually 10%;
- As an ethyl tertiary butyl ether (ETBE) component of reformulated gasoline both directly and/or transformed into a compound.

Using ethanol as gasohol or ETBE does not require specially configured vehicles. Almost all current vehicles can work with these fuels without a problem, with likely favourable emissions. The Environmental Protection Agency (EPA) has refined ethanol and ETBE mixtures in gasoline for mandatory use during wintertime, which aims to reduce the vehicle's carbon monoxide. The blend E85 is characterized by a low sulphur content. This reduces catalyst deterioration compared to that of vehicles working with gasoline.

The combustion properties of ethanol, such as flash point and the auto-ignition temperature, are higher than those of combustion gasoline, which makes it safer for storage and transportation. The latent heat of evaporation of gasoline is between 3 and 5 times

lower than that of ethanol. When using ethanol in ICE, the volume efficiency increases because the temperature of the intake manifold is lower. Compared to gasoline, ethanol has a lower heating value, and as consequence of this, more alcohol fuel is required (about 1.6 times) to achieve the same output power. The ethanol stoichiometric air–fuel ratio (Table 1) is lower than that of gasoline (about 2/3), therefore, the required air for complete combustion is less for ethanol [34]. The lower vapor pressure of ethanol (Table 1) results in fewer evaporative emissions. The simple structure of the ethanol molecule makes it an appropriate fuel for gasoline engines. The ethanol is characterized by a high-octane number; this allows for higher compression compared to that of gasoline [35]. The higher-octane number permits the mitigation of the knocking phenomena, allowing the usage of a higher compression ratio. Many researchers have studied gasoline engine performance with ethanol blends. These investigations have found that the use of ethanol/gasoline blends with low concentrations of ethanol (<20% by volume) have little influence on the engine power and torque [36–38]. Storey et al. [39] studied and compared PM emissions of GDI engines (using lean and stoichiometric mixtures) with E0, E10, and E20 ethanol/gasoline blends. The experimental results showed a reduction of PM mass emissions (between E0 and E10 it is 29% in the stoichiometric mixture, and a 42% reduction was seen in the lean mixture) under the Federal Test Procedure 75 (FTP). Vertin et al. [40] studied the effects of ethanol/gasoline blending on emissions. The experimental results showed that E15 or E20 fuel had lower exhaust emissions compared to those of vehicles using E0 fuel. Jung et al. [41] studied gasoline engines using ethanol compared to gasoline at part loads and reported a 25%–45% decrease in NOx for E85. Schifter et al. [42] investigated the influence of E17–E24 (17–24% ethanol) ethanol/gasoline fuel blends (0–20% ethanol) on emissions and performance. The experiment showed decreased CO and HC emissions and increased NOx emissions when using an E20 blend.

### 1.3. Properties of Butanol

Butanol is a promising alcohol fuel that is another candidate as an alternative fuel. It can be used with gasoline in ICE without engine or fuel system modification. There are two ways to produce butanol: One way is by using fossil fuels, and this butanol is known as petrobutanol. The other way is to produce it is by using biomass, and this butanol is known as biobutanol. The chemical properties of the two butanols are the same. Compared to ethanol and methanol, butanol is less corrosive to metal and rubber because it is less hygroscopic. It is also less prone to water contamination and thus could be transported more easily than gasoline can be. Compared to ethanol and methanol, the combustion properties of butanol are closer to those of gasoline. Butanol has better cold start properties than does ethanol and can burn at a wider temperature range. In addition, butanol has lower vapor pressure and a high enough octane number, close to that of gasoline. Compared to ethanol and methanol, the octane number of n-butanol is lower but similar to that of gasoline (Table 1). The butanol's anti-knock properties are similar to those of gasoline due to the octane number close to that of gasoline [43]. The mentioned properties of butanol make it more suitable than other alcohols (ethanol and methanol) for blending with gasoline. From Table 1, it can be seen that butanol has the lowest heat of evaporation, compared to that of ethanol and methanol. Fuels with a higher heat of evaporation are suitable for engines with PFI (port fuel injection) systems, because they cause lower intake charge temperature and complete vaporization in the intake port. This increases the mass of the charge and density of the combustible mixture. Due to a higher laminar speed of butanol flame propagation than that of gasoline, the combustion process completes earlier, and the heat efficiency of the engine is improved. Compared to ethanol and methanol, the heating value of n-butanol is higher (see Table 1), which is less likely to affect the fuel consumption, but it will improve fuel economy. Many studies have been conducted on different alcohol blends of methanol, ethanol, butanol, and gasoline in various spark-ignition engines. In one of the studies [44], the authors conducted research on the influence of 5% and 10% methanol and ethanol blending in gasoline on

engine performance and emissions. The obtained experimental results showed that blends (M10 and E10) reduced HC emissions by 13% for methanol blends and 15% for ethanol blends, reduced CO emissions by 10.6% for methanol blends and 9.8% ethanol blends, and increased $CO_2$ emissions for both the methanol and ethanol blends compared with those of gasoline. The blended fuels M10 and E10 showed an increase in BSFC (brake specific fuel consumption) and a decrease in brake thermal efficiency in comparison to those of gasoline. The authors of another study [29] used a three-cylinder gasoline engine with PFI to study torque, power, specific fuel consumption, emissions, and cold start characteristics with methanol/gasoline fuel blends. Those experiments showed that the engine power and torque were slightly lower when using a methanol/gasoline blend, while the blends increased engine brake thermal efficiency (BTE). In addition, methanol-blended fuel lowered HC and CO emissions, and blended fuel improved the cold start. In another study [45], the authors conducted an experiment with a PFI engine with increasing concentrations of n-butanol fuel blends and pure n-butanol. During the study, it was found that HC emissions increased as n-butanol increased, the emissions of CO did not show a significant change, and the effects on NOx were small and inconsistent. Wallner et al. [46] conducted experimental work with net gasoline and blends E10 (ethanol 10%) and B10 (butanol 10%) in a four-cylinder GDI engine and investigated the NOx, HC, and CO emissions. The results showed that there was a small difference in emissions (NOx, HC, and CO) when running the engine with B10 and net gasoline. The reason for this is that the engine operates with a stoichiometric air–fuel ratio for net gasoline, E10, and B10, thus, an excess of oxygen is not available.

In the review of the literature, there is not enough information about the comparison of methanol, ethanol, and n-butanol blending with gasoline and its effect on engine performance and emissions in the same PFI SI engine and under the same conditions. A comparison between blended fuels and net gasoline in spark ignition engines is important to understand which kind of blended fuel is more appropriate for reducing exhaust emissions of an SI engine. It is also important to understand their impact on torque, power, and fuel consumption. Therefore, the objective of this work was to compare ethanol, methanol, and butanol blending with gasoline and its effect on engine performance and emissions without modification on ICE.

## 2. Materials and Methods

*Simulation Setup*

Developments in computational models and market penetration of high-performance computers have allowed for the simulation and analysis of various processes regarding the performance of ICE. Today, the advanced simulation tools used in internal combustion engines make it possible to investigate how to achieve high engine efficiency and low fuel consumption and emissions while maintaining good performance based on typical operating conditions. In the simulation of ICE, models of varying complexity are used, ranging from the simpler approach of the 1D (one-dimensional) model to more complex levels of detail iin the 3D models. There are different types of simulation models depending on their complexity: 0D zero-dimensional, 1D one-dimensional, and 3D three-dimensional. The 0D and 1D models are single zone and are the least complex, while 3D models are multi-zone and multidimensional models, and they are the most complex due to the need for more details [47,48].

The present research aimed to create a 1D model of an engine with PFI to study the influence of blends of ethanol, methanol, and butanol with gasoline on the emitted emissions and engine power. The one-dimensional model was developed with the help of a specialized software product AVL BOOST. The gasoline engine model was calibrated and described by Iliev [49] and its layout is shown in Figure 1 with a library of the used elements in Table 2 and engine specifications shown in Table 3.

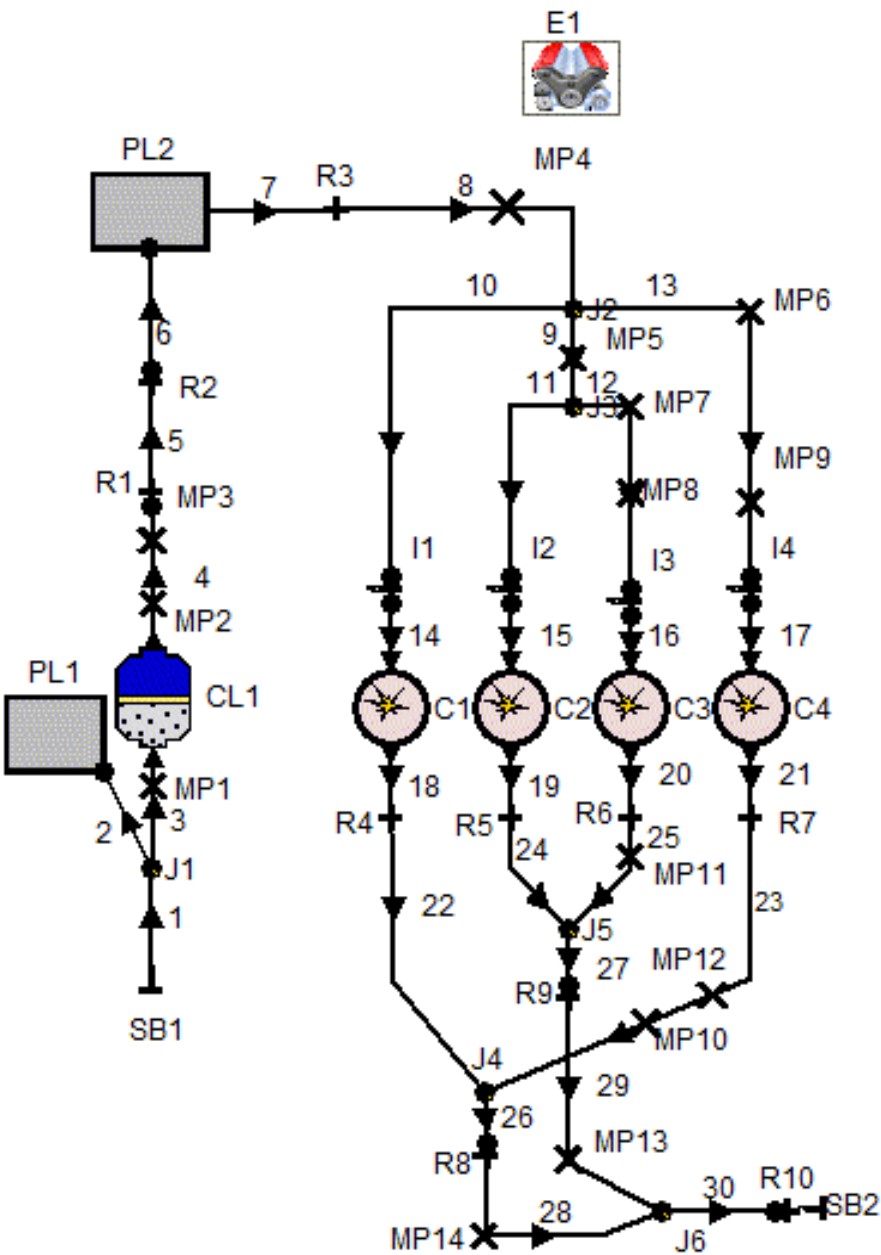

**Figure 1.** Layout of the PFI engine model.

**Table 2.** Library with the used elements.

| Elements | Description |
| --- | --- |
| E1 | Engine management |
| C1, C2, C3, and C4 | Number of engine cylinders |
| MP1 to MP18 | Measuring points |
| PL1, PL2, PL3, and PL4 | Plenums |
| SB1 and SB2 | System boundaries |
| 1 to 30 | Flow pipes |
| CL1 | Cleaner |
| R1 to R10 | Flow restrictions |
| I1 to I4 | Fuel injectors |

**Table 3.** Gasoline PFI engine specifications.

| Engine Parameters | Value |
|---|---|
| Bore | 86 (mm) |
| Stroke | 86 (mm) |
| Compression ratio | 10.5 |
| Connection rod length | 143.5 (mm) |
| Number of cylinders | 4 |
| Piston pin offset | 0 (mm) |
| Displacement | 2000 (cc) |
| Intake valve open | 20 BTDC (deg) |
| Intake valve close | 70 ABDC (deg) |
| Exhaust valve open | 50 BBDC (deg) |
| Exhaust valve close | 30 ATDC (deg) |
| Piston surface area | 5809 (mm2) |
| Cylinder surface area | 7550 (mm2) |
| Number of strokes | 4 |

In this research, a two-zone model of Vibe was chosen for the combustion simulation and analysis. The combustion chamber was divided into unburned and burned gas regions [50]. However, the assumption that burned and unburned charges have the same temperature was dropped. Instead, the first law of thermodynamics was applied to both the burned charge and unburned charge.

$$\frac{dm_b u_b}{d\alpha} = -p_c \frac{dV_b}{d\alpha} + \frac{dQ_F}{d\alpha} - \sum \frac{dQ_{Wb}}{d\alpha} + h_u \frac{dm_b}{d\alpha} - h_{BB,\,b} \frac{dm_{BB,b}}{d\alpha} \qquad (1)$$

$$\frac{dm_u u_u}{d\alpha} = -p_c \frac{dV_u}{d\alpha} - \sum \frac{dQ_{Wu}}{d\alpha} + h_u \frac{dm_b}{d\alpha} - h_{BB,\,u} \frac{dm_{BB,u}}{d\alpha} \qquad (2)$$

where $dm_u$ represents the change of the internal energy in the cylinder, $p_c \frac{dV}{da}$ is the piston work, $\frac{dQ_F}{da}$ stands for the fuel heat input, $\frac{dQ_W}{da}$ is wall heat loses, and $h_u \frac{dm_b}{da}$ represents the enthalpy flow from the unburned to the burned zone due to the conversion of a fresh charge to combustion products. Heat flux between the two zones is neglected. $h_{BB} \frac{dm_{BB}}{da}$ is the enthalpy due to blow by, u and b in subscript are unburned and burned gas, respectively. Moreover, the sum of the volume changes must be equal to the cylinder volume change and the sum of the zone volumes must be equal to the cylinder volume.

$$\frac{dV_b}{d\alpha} + \frac{dV_u}{d\alpha} = \frac{dV_u}{d\alpha} \qquad (3)$$

$$V_b + V_u = V \qquad (4)$$

The amount of burned mixture at each time setup is obtained from the Vibe function. For all other terms, for instance wall heat losses etc., models similar to the single zone models with an appropriate distribution on the two zones were used [51].

The engine specifications and library with the used elements are shown in Tables 2 and 3, respectively.

## 3. Results

The present research was based on a study of performance and emission characteristics of a PFI gasoline engine working with blends of methanol, ethanol, and butanol with net gasoline using AVL BOOST software. The experiments were conducted at full load for the engine speeds from 1000 to 6500 rpm. The following blends were used: 0% methanol (ethanol, butanol) M0 (E0, B0), 5% methanol (ethanol, butanol) M5 (E5, B5), 10% methanol (ethanol, butanol) M10 (E10, B10), 20% methanol (ethanol, butanol) M20 (E20, B20), 30%

methanol (ethanol, butanol) M30 (E30, B30), 50% methanol (ethanol, butanol) M50 (E50, B50), and 85% methanol (ethanol, butanol) M85 (E85, B85) by volume.

### 3.1. Engine Performance Characteristics

The results for brake power obtained by blended alcohols and net gasoline are shown in Figure 2.

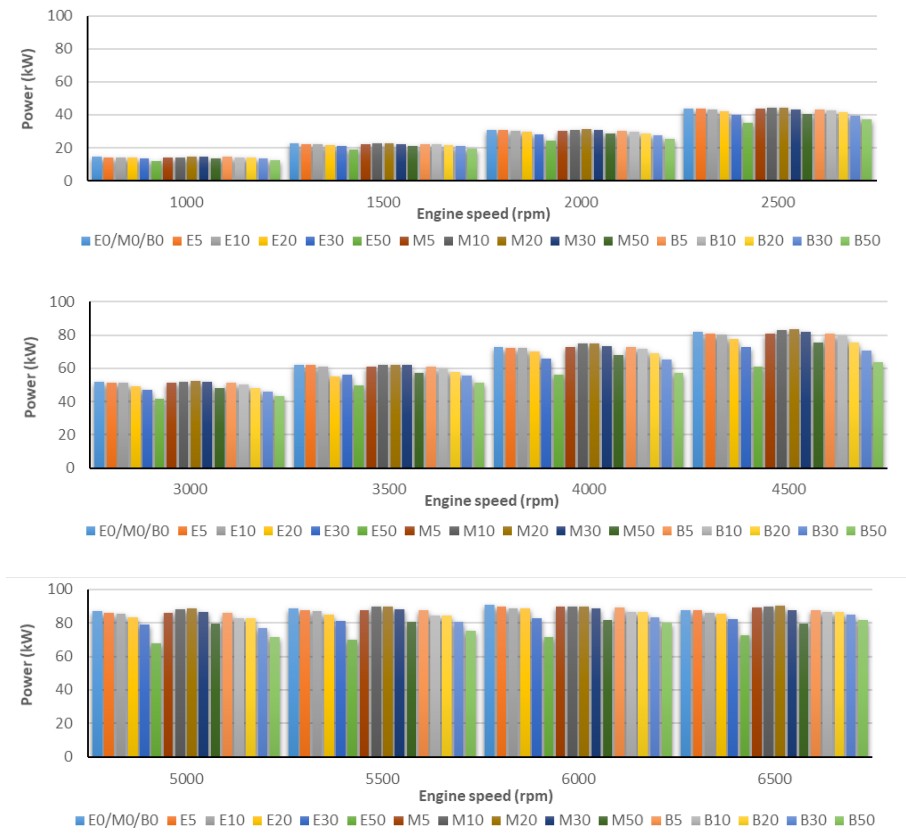

**Figure 2.** The results for engine brake power obtained from blended alcohol fuels and net gasoline.

It can be observed that the engine brake power decreased with increasing ethanol in the fuel blends. The highest power was obtained when the engine was running on net gasoline. The reason for this phenomenon is due to the heating value. Ethanol is characterized by a lower heating value than that of net gasoline (Table 1) and with an increase in ethanol in the fuel blends, the heating value of this blend also decreases. As a result, engine brake power decreases [49].

Similar results were observed when operating the engine with methanol blends. The brake power slightly increased with increasing methanol in the fuel blends (M5 and M10). The obtained result is due to the better combustion efficiency of alcoholic (oxygenated) fuel. It was noticed that with increased methanol in the blends (M30 and M50), the power decreased, which is due to the lower heating value of methanol relative to that of gasoline (Table 1). This effect was observed over the entire range of engine speeds and in all mixtures with butanol.

At low engine speeds, the addition of butanol to gasoline did not significantly affect brake power. When the engine speed was increased above 4000 min$^{-1}$, a significant influence of n-butanol on the engine power was observed (see Figure 2). The possible reason for this effect of butanol can be explained by the lower calorific value of n-butanol compared to that of net gasoline. It should also be borne in mind that the latent heat of n-butanol (84 kJ/kg) is higher than that of gasoline (349 kJ/kg). It follows that n-butanol absorbs more heat to evaporate and burn.

The results for brake specific fuel consumption (BSFC) obtained by blended alcohols and net gasoline are shown on Figure 3. The results were obtained at full load and various engine speeds. It can be observed that the BSFC increased with increasing alcohol content (methanol, ethanol, and n-butanol) in the fuel blends. The obtained results are due to the lower heating value and stoichiometric air–fuel ratio of alcohols compared to those of net gasoline, which leads to the need for more fuel for that specific air–fuel equivalence ratio.

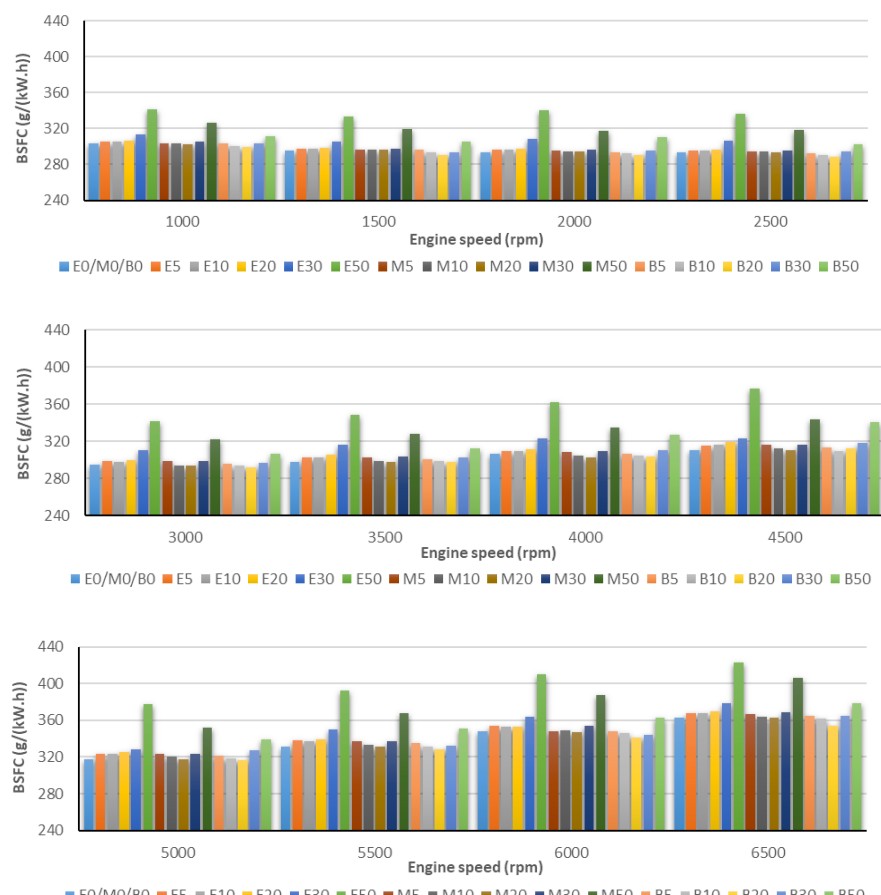

**Figure 3.** The results for BSFC obtained from blended alcohol fuels and net gasoline.

In addition, there was a small difference between the BSFC when the engine was running with net gasoline and blended fuels (E5 (M5, B5), E10 (M10, B5), and E20 (M20, B5). The obtained results were due to the lower energy content of blended alcohol fuels caused by a small difference between the BSFC.

*3.2. Emissions Characteristics*

The comparison of CO emissions for blended fuels of ethanol, methanol, and butanol with net gasoline at engine speeds from 1000 min$^{-1}$ to 6500 min$^{-1}$ is shown in Figure 4.

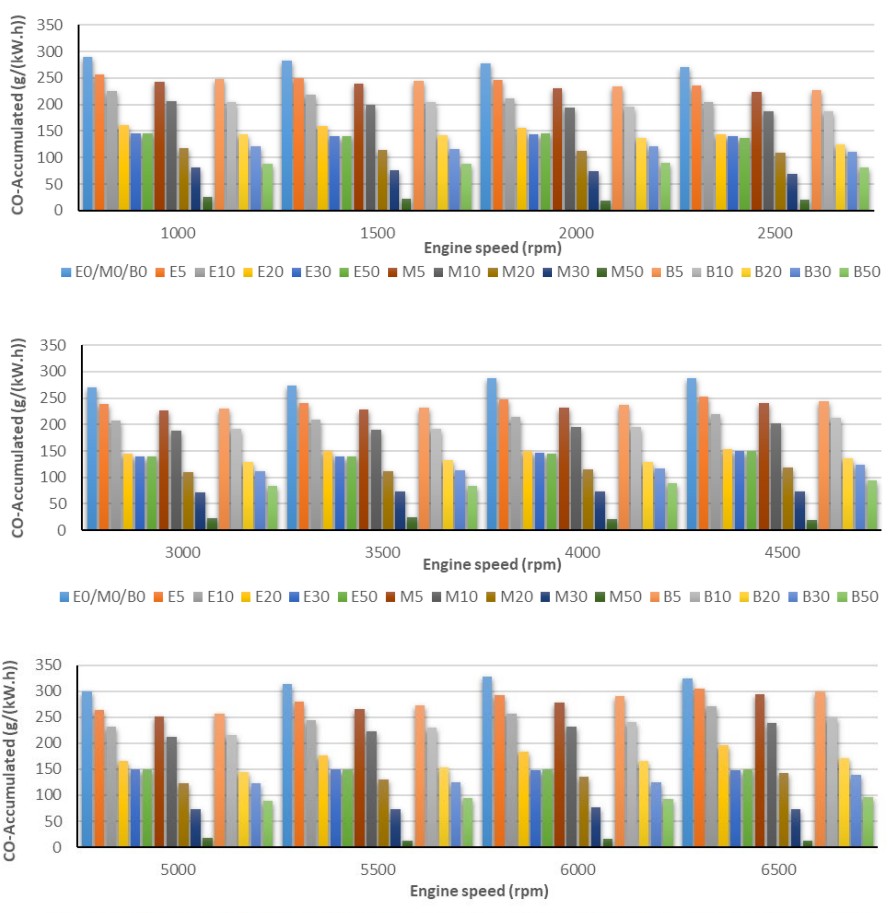

**Figure 4.** The results for CO emissions obtained from blended alcohol fuels and net gasoline.

It can be observed that as the ethanol, methanol, and butanol percentages in blends increased, the CO emissions decreased compared with that of net gasoline. The reason for the formation of CO is due to incomplete combustion due to lack of oxygen in the fuel mixture and due to insufficient time to complete the combustion process. With the improvement of the combustion process due to the use of oxygen-containing additives such as methanol, ethanol, and butanol there is a reduction in the formation of CO emissions. Finally, ethanol ($C_2H_5OH$), methanol ($CH_3OH$), and butanol ($C_4H_9OH$) have less carbon than does gasoline ($C_8H_{18}$), which is also a reason for the reduced CO emissions. This has been reported by other authors [52,53].

The comparison of HC emissions for blended fuels of ethanol, methanol, and butanol with net gasoline at engine speeds from 1000 $min^{-1}$ to 6500 $min^{-1}$ is shown in Figure 5. It can be observed that as the ethanol, methanol, and butanol percentages in blends increased, the HC emissions decreased compared with that of net gasoline. The reason for this phenomenon is due to the same reasons as those for the decrease in CO emissions described in the previous paragraph.

In addition, as the relative air–fuel ratio increased, HC emissions decreased.

Based on the obtained results for HC emissions, it was found that methanol was the most suitable fuel compared to ethanol. Lower HC emissions were found with more complete combustion processes. Other authors have reached the same conclusion [40,41].

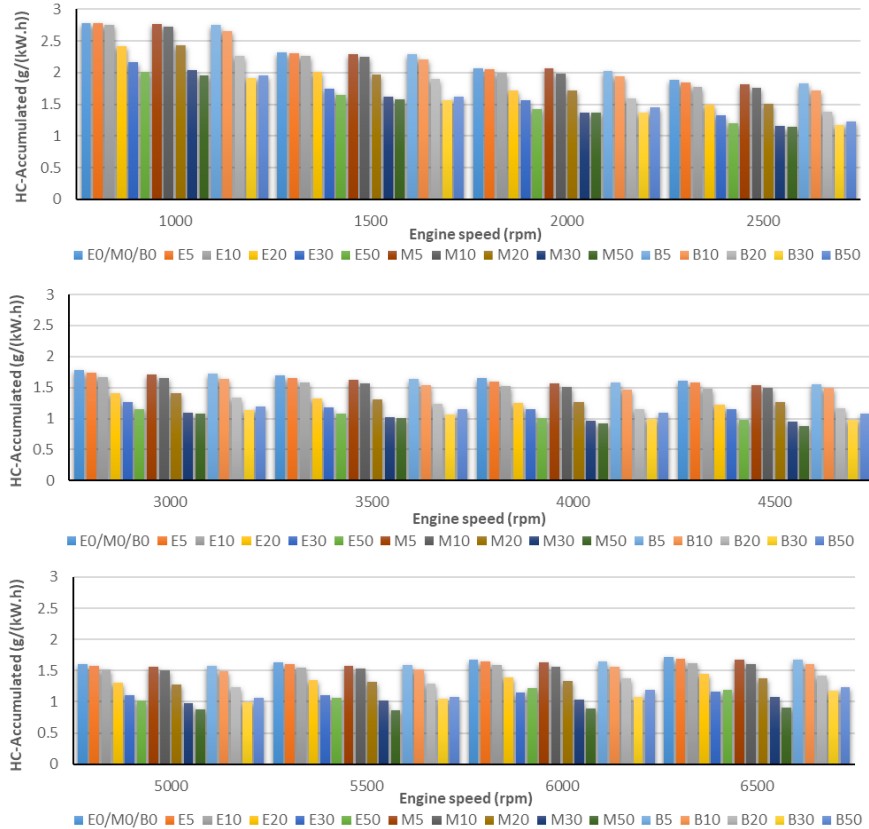

**Figure 5.** The results for HC emissions obtained by blended alcohols fuels and net gasoline.

The comparison of NOx emissions for blended fuels of ethanol, methanol, and n-butanol with net gasoline at engine speeds from 1000 min$^{-1}$ to 6500 min$^{-1}$ is shown in Figure 6. Nitrogen oxides NOx are formed during the oxidation of nitrogen from the air during combustion. There are two important factors for the formation of nitrogen oxides: high temperature and the presence of free oxygen [54]. Therefore, areas with very high temperatures are a likely source of nitrogen oxide formation.

It can be observed that when the ethanol and methanol percentage in blends increased up to 30% E30 (M30), the NOx emissions increased, after which with further increase of ethanol (methanol), the NOx emissions decreased. The n-butanol showed an increase in NOx emissions when the n-butanol percentage in blends increased up to 50% (B50), after which with further increases of n-butanol, the NOx emissions decreased. The reason for the increase in emissions can be explained by the improvement of the combustion process, as a result of which the temperature in the engine cylinder also increases.

The reason for the reduced NOx emissions at high percentages of alcohol in the mixtures is due to the reduction of the temperature in the cylinder. The reduced temperature is due to 1. the latent heat of vaporization of methanol and n-butanol; their evaporation reduces the temperature in the cylinder, and 2. the fact that as more triatomic molecules are produced, the gas heat capacity increases and the combustion gas temperature decreases. However, the low in-cylinder temperature can also lead to an increment in the unburned combustion product.

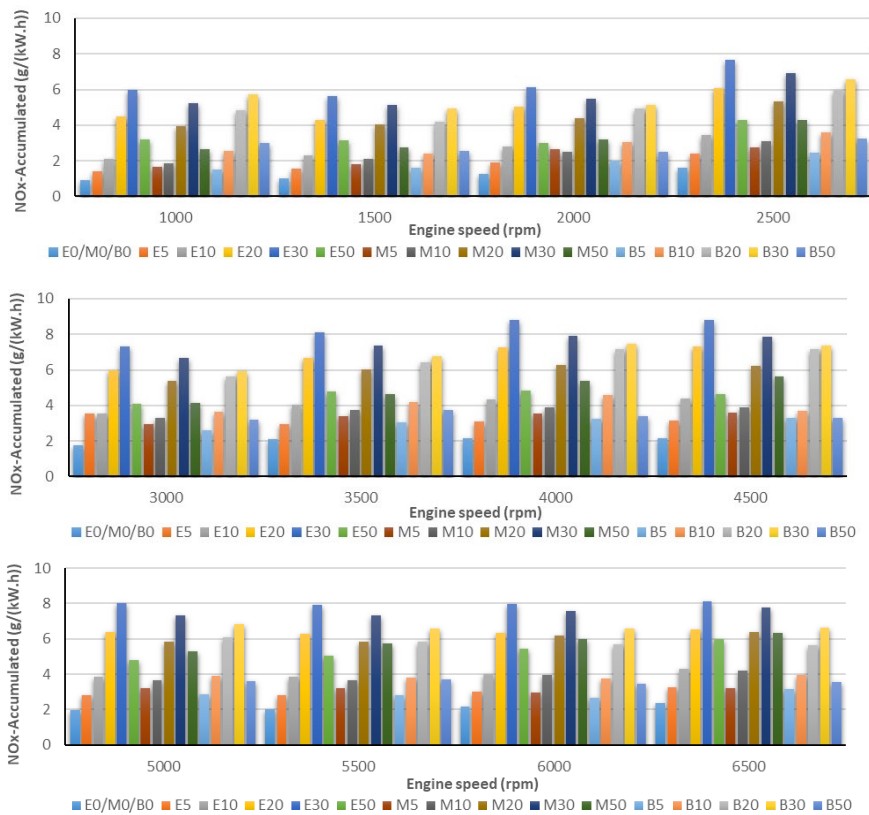

**Figure 6.** The results for NOx emissions obtained from blended alcohol fuels and net gasoline.

## 4. Conclusions

This paper discusses the influence of blends of gasoline with ethanol, methanol, and butanol on the operation of a gasoline engine. The effect of gasoline/alcohol blends on engine performance and emissions were studied. The results of this study can be summarized as follows:

1.  The engine brake power decreased with increasing ethanol and methanol in the fuel blends. The brake power slightly increased with increasing methanol in the fuel blends (M5 and M10). It was noticed that with an increase of methanol in the blends (M30 and M50), the power decreased;

2.  The BSFC increased with increasing alcohol content (methanol, ethanol, and n-butanol) in the fuel blends. There was a small difference between the BSFC when the engine was running with net gasoline and blended fuels E5 (M5, B5), E10 (M10, B5), and E20 (M20, B5);

3.  When ethanol, methanol, and butanol percentage in blends increased, the CO emissions decreased compared with that of net gasoline;

4.  When ethanol, methanol, and butanol percentage in blends increased, the HC emissions decreases compared with that of net gasoline. Based on the obtained results about HC emissions, it was found that methanol was the more suitable fuel compared to ethanol;

5.  When ethanol and methanol percentage in blends increased up to 30% E30 (M30), the NOx emissions increased, after which, with a further increase of ethanol (methanol) the NOx emissions decreased. The n-butanol showed an increase in NOx emissions when n-butanol percentage in blends increased up to 50% (B50), after which, with a further increase of n-butanol, the NOx emissions decreased.

**Funding:** This research received no external funding.

**Acknowledgments:** The present document was produced with the financial assistance of the Project 2021-RU-03 "Investigation the possibilities for optimizing the energy consumption of an electric car of the prototype class for the Shell Eco-marathon race". We are also eternally grateful to AVL-AST, Graz, Austria for granting use of the AVL-BOOST under the university partnership program.

**Conflicts of Interest:** The author declares no conflict of interest.

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
