# Peer review of "A Comparison of Ethanol, Methanol, and Butanol Blending with Gasoline and Its Effect on Engine Performance and Emissions Using Engine Simulation"

_processes, doi:10.3390/pr9081322_

Round 1
Reviewer 1 Report
The article deals with an important and topical topic. It can contribute to the development of the field of science and technology that he studies.
The article has a non-classical structure (introduction, methodology and material, research, analysis, summaries), which makes the introduction look short and poorly recognized in the literature. However, reading further, the author describes the selected fuels in detail.
The analysis of the results is clear despite the very large number of results.
Detailed Notes:
This statement should be supported by a literature source: „The depletion of crude oil is a problem that has arisen in recent decades.”
The author of his statements in the article should confirm with scientific articles in which selected issues are described, e.g.
- „Numerous studies have been conducted in recent years to find substitutes for fossil fuels.” – current articles showing the search for such fuels should be cited.
- “Another very important problem is the combustion gases emitted from ICE (internal combustion engines) that negatively influence of the nature and people life.” – articles showing the negative impact of exhaust gases on human life and health, e.g. doi :10.1088/1755-1315/90/1/012036 ,
- “The harmful exhaust gases, released from ICE, are hydrocarbons (HC), nitrogen oxides (NOx), carbon monoxide (CO), and pollutants from particles.” Are these really all ingredients? Or are they the main components of exhaust gases? It is worth referring to literature.
- „The CO2 emissions are estimated as a pollutant because they cause of global warming. Also, CO2 is one of the gases responsible for the greenhouse gas emissions (GHGE).” Agreed, but support it with literature. In addition, CO2 is dangerous for people because it takes the place of oxygen in the air that people breathe, it is not toxic, but causes hypoxia in the body described, e.g. https://doi.org/10.3390/polym12102232 , https://doi.org/10.1038/s41598-021-87634-9 , DOI: 10.15199/62.2020.1.12 .
- „Alternative fuels or energy sources are obtained from resources other than petroleum which can contribute to the decarbonisation of transport and improve the environmental perfor mance of the transport sector.” It is worth referring to work on such fuels or using methods of introducing admixtures from such fuels: https://doi.org/10.3390/en14133939 , https://doi.org/10.3390/en14113077 , https://doi.org/10.3390/app11041411 .
- „The fuels that are most often used as alternative fuels are propane, natural gas, methanol, ethanol, butanol and hydrogen.” This sentence should also be supported by current articles showing the introduction and research on the alternative fuels mentioned. Liquefied petroleum gas (LPG) should also be replaced. “Natural gas (NG) is an alternative fuel within the meaning of the European Union Directive (2014/94/UE), as it is an alternative for energy sources derived from crude oil [1]”, https://doi.org/10.3390/en13246709 .
[1] EU. Directive 2014/94/EU of the European Parliament and of the Council of 22 October 2014 on the Deploy-Ment of Alternative Fuels Infrastructure Text with EEA Relevance; EU: Strasbourg, France, 2014.
“Liquefied petroleum gas (LPG) is an alternative fuel within the meaning of the European Union Directive (2014/94/UE), as it is an alternative for energy sources derived from crude oil.” https://doi.org/10.3390/en13215773 .
The sentence should be changed and supported by citations, e.g. „The fuels that are most often used as alternative fuels are propane [], natural gas [ https://doi.org/10.3390/en13246709 ], liquefied petroleum gas [ https://doi.org/10.3390/en13215773 , https://doi.org/10.3390/en13112995 ], methanol [https://doi.org/10.1016/j.fuel.2019.116403], ethanol [https://doi.org/10.1007/s11356-017-9651-8 , https://doi.org/10.1007/s11356-018-2476-2], butanol [] and hydrogen [].”
- “ Many of these fuels are used as additives to fossil fuels (gasoline and diesel 42 fuels) and can be blended with them. Fuel additives can be added to the base fuel to in prove its properties.” cite works describing such mixtures, e.g. https://doi.org/10.3390/app10010359
Line 58 shouldn't there be NOx instead of NO?
The quoted values of 94% do not come from the results of the research described in the article, so they should be referred to the literature. „At the same fuel efficiency, the CO2 emissions emitted by vehicles working with methanol are theoretically around 94% of the emissions of similar vehicles with petroleum.”
I will read the revised version with curiosity. Maybe the author will come up with some additional way of presenting the graphical statement of the results.
Author Response
Dear Reviewer,
I have made corrections to all the remarks made.
I could not find another additional way to present the graphical statement of the results. Due to the large number of results obtained, it is difficult to visualize them in an appropriate way.
Thanks for the comments!

Reviewer 2 Report
- Section “Abstract” should be corrected. Please delate general phrases such as “Air pollution, especially in large cities around the world, is associated with serious problems both with people’s health and the environment. Although, over the past few years, there has been a particularly intensive demand for alternatives to fossil fuels because when they are burned, substances that pollute the environment are released. In addition to the smoke from fuels burned for heating and harmful emissions that industrial installations release, the exhaust emissions of vehicles have a large share. Alternative fuels, known as non-conventional and advanced fuels are derived from resources other than fossil fuels. Because alcoholic fuels have several physical and fuel properties similar to gasoline, they can be considered as one of the alternative fuels.”. Please focus on the obtained results.
- There is no analysis of previously obtained results in the section “1. Introduction”. Critical analysis should show the novelty of the present study. Please add the purpose of the study at the end of the section “1. Introduction”.
- Sections “2. Properties of Methanol”, “3. Properties of Ethanol”, “4. Properties of Butanol” should be combined in one section “2. Materials”. Please avoid using general well-known information.
- Section “5. Simulation Setup”. There is no information about mathematical model, numerical simulation algorithm and numerical simulation methods. Verification of mathematical model and numerical simulation results should be added.
- Section “6. Results” does not contain scientific results. The simple description of the figures is given in the text. For example it would be better to present mechanisms (these may be mechanisms previously developed by other authors) that explain the difference in emissions for different fuels in the section “6.2. Emissions Characteristics”.
Author Response
Dear Reviewer,
I am sending answers to your review
Point 1: Opinions to the two reviewers are different. That's why I've added a lot of supporting literature. The paper has the purpose of the study at the end of the section “1. Introduction”.
Point 2: Sections “2. Properties of Methanol”, “3. Properties of Ethanol”, “4. Properties of Butanol” were combined in one section “1. Introduction. I have reviewed similar published articles and considered that section 1 is the appropriate place for these sections
Поинт 3: Сецтион 5 I have already published articles describing the mathematical model and its validation, so I do not want to repeat things that have already been published. That's why I've added the basic equations used in the model.
Point 3 Section “6 If I start describing the mechanisms of emission formation, it will turn out so that I use well-known information.
Thanks for the review!

Round 2
Reviewer 1 Report
The article has been corrected.
Reviewer 2 Report
Please add recomendations for practical application from obtained results at the end of the manuscript.